# A Review on Therapeutic Potential of Natural Phytocompounds for Stroke

**DOI:** 10.3390/biomedicines10102566

**Published:** 2022-10-13

**Authors:** Farooq M. Almutairi, Aman Ullah, Yusuf S. Althobaiti, Hafiz Muhammad Irfan, Usman Shareef, Halima Usman, Sagheer Ahmed

**Affiliations:** 1Department of Clinical Laboratories Sciences, College of Applied Medical Sciences, University of Hafr Al-Batin, Hafr Al-Batin 39524, Saudi Arabia; 2Saba Medical Center, Abu Dhabi P.O. Box 20316, United Arab Emirates; 3Department of Pharmacology and Toxicology, College of Pharmacy, Taif University, Taif 21944, Saudi Arabia; 4Addiction and Neuroscience Research Unit, Taif University, Taif 21944, Saudi Arabia; 5College of Pharmacy, University of Sargodha, Sargodha 40100, Pakistan; 6College of Pharmaceutical Sciences, Shifa Tameer-e-Millat University, Islamabad 44000, Pakistan

**Keywords:** stroke, phytocompounds, occlusion of blood, paralysis, ischemia

## Abstract

Stroke is a serious condition that results from an occlusion of blood vessels that leads to brain damage. Globally, it is the second highest cause of death, and deaths from strokes are higher in older people than in the young. There is a higher rate of cases in urban areas compared to rural due to lifestyle, food, and pollution. There is no effective single medicine for the treatment of stroke due to the multiple causes of strokes. Thrombolytic agents, such as alteplase, are the main treatment for thrombolysis, while multiple types of surgeries, such ascraniotomy, thrombectomy, carotid endarterectomy, and hydrocephalus, can be performed for various forms of stroke. In this review, we discuss some promising phytocompounds, such as flavone C-glycoside (apigenin-8-C-β-D-glucopyranoside), eriodictyol, rosamirinic acid, 6″-*O*-succinylapigenin, and allicin, that show effectiveness against stroke. Future study paths are given, as well as suggestions for expanding the use of medicinal plants and their formulations for stroke prevention.

## 1. Introduction

Stroke is a devastating medical condition and a frequent cause of disability and mortality globally [1]. The prevalence of this disease is more pronounced in underdeveloped countries than in developed countries. Since the size of the aged population is constantly increasing, there has been an upsurge in the incidence of stroke globally. Epidemiological data indicate that almost 1.1% of the inhabitants of Europe suffer from stroke per annum. Moreover, its prevalence is 1.5 times higher in males than females [2,3]. In addition, the prevalence of ischemic stroke in young adults is increasing, implying the need for targeted preventive measures in that age group [4,5]. Sufferers of stroke experience severe neurological disabilities, accompanied by recurrent hospitalization and other complications, such as venous thromboembolism, syncope, bone fractures, and infections. Long-term nursing care and rehabilitation are costly, and stroke survivors and their caregivers face a substantial financial burden [6,7]. 

Ischemic stroke constitutes 85% of all stroke cases, while the remaining cases are haemorrhagic (10%) and subarachnoid stroke (3%) [8]. Ischemic stroke results from the occlusion of an artery leading to a particular area of the brain that limits the oxygen supply and, therefore, results in neuronal cell death (Figure 1), while haemorrhagic stroke is caused by aberrant vascular structure or the rupture of blood vessels [9]. Stroke has complex pathophysiology involving a cascade of pathological events, such as free radical generation, inflammation, perturbed ionic homeostasis, excitotoxicity, and ultimately irreversible neuronal damage and apoptosis [10,11].

The only medication for post-stroke treatment that the FDA has approved is tissue plasminogen activator. However, fewer patients are eligible for its use due to its time-restricted benefits and associated adverse drug reactions. Several trials conducted for novel drug discovery for stroke have led nowhere. Researchers are yet to achieve success in finding alternate neuroprotective therapy for stroke. These repeated failures have diverted researchers’ attention towards using phytochemicals in the treatment of ischemic stroke. Many convincing studies show diverse beneficial effects of phytochemical use against various pathologies involved in ischemic strokes, such as inflammation, oxidative stress, and apoptosis [5]. This review focuses on the promising phytochemical compounds identified from pre-clinical studies that can be used either independently or in conjunction with already existing stroke treatments.

## 2. Pathophysiology

The pathophysiology of stroke needs to be adequately understood to avert mortality and disability. Stroke results from reduced blood flow to the brain [12]. There are two main types of strokes, haemorrhagic and ischemic. Of these, ischemic stroke is the most prevalent [13] and is the type mainly addressed in this review. 

Ischemic stroke is the consequence of reduced blood supply due to occlusion of the main cerebral artery, either temporarily or permanently. The circulation of blood after occlusion does not remain uniform. The central cerebral core is devoid of blood supply, leading to rapid neuronal damage [5,13] due to lipolysis, proteolysis, and degradation of microtubules [14]. The supply of blood flow to the core falls below 5–8ml/100g/minute [13], and ionic homeostasis is lost [14]. The area surrounding the core where functional activity is lost, but cell death has not yet occurred is known as the penumbra [12]. This zone is unstable and can be restored if timely reperfused. It is a small area and exists for a shorter period. The penumbra is surrounded by a zone known as the oligemic compartment. This area has little hypoperfusion and normal oxygen consumption. If the blood supply obstruction is not restored, the infarct area may continue to expand [13]. Additionally, as depolarization of neuronal and glial cells increases, the infarct area enlarges.

The cascade of events following ischemic stroke leads to neuronal cell death. As the blood flow to the area of insult is diminished, it may result in hypoxia, decreased ATP production, lactic acidosis, and disturbance of homeostasis. The brain needs a large amount of oxygen and glucose to produce energy. These substrates are supplied through blood. Energy production is halted on the occlusion of blood supply [14]. The functional activity of energy is driven by the Na/ K ATPase pump, and the plasma membrane Ca ATPase pump is lost, and there occurs a marked increase in intracellular concentration of calcium [15].

An increased intracellular Ca concentration causes the vesicles to fuse with the vesicular membrane and release excitatory neurotransmitters such as glutamate. Additionally, the reuptake of neurotransmitters is reduced [14]. Hyperstimulation due to increased glutamate results in an increased influx of Ca and an increased level of Na and Cl ions in neurons. Water moves in the direction of Na and Cl, resulting in edema. Overloading calcium activates proteases, lipases, and nucleases, triggering various catabolic processes: proteases damage cell membranes and proteins. The increase in ROS and reactive nitrogen species due to excessive Ca also led to cell swelling and death [14].

Hyper-depolarization of the mitochondrial membrane occurs due to excessive calcium intake that disrupts mitochondrial function. This downstream cell death pathway is due to free-radical formation and enzyme and protease release. Mitochondria release cytochrome C through mitochondrial transition permeability pores that activate caspase causing DNA breakdown and activating apoptosis [14]. Furthermore, free radical formation, lipoxygenase cascade, and disruption of ionic balance lead to cell death pathways [12]. BCL-2 in mitochondria is the protein that mainly regulates cell death. Bax and Bok are pro-apoptotic proteins. After the pro-apoptotic proteins are released into the cytoplasm, they activate a cascade of caspases. These break down proteins and enzymes, disturb the cell’s homeostasis, and lead to cell death [15].

Blood supply to the ischemic core after stroke is significantly reduced. As a result, ATP production also declines. Disruption of the Na/K ATPase and Ca ATPase pumps occurs. The concentration of Na and K becomes dysregulated. Accumulating Na inside the cell causes swelling that may cause the cell to rupture and necrosis. Cell rupture releases cellular components and initiates an inflammatory process.

The increased intracellular concentration of Ca and hypoxia produce free radicals and initiate inflammation and apoptosis. This releases nitric oxide synthase, which is responsible for tissue damage. Ischemic strokes show vascular response through the initiation of inflammation. TNF α, platelet-activating factor, and IL β are produced due to injury to the brain cells, and adhesion molecules appear on endothelial cells. These adhesion molecules cause the neutrophils to stick to them. Monocytes and macrophages follow the neutrophils. Cells of the brain, such as astrocytes and microglia, also become activated [14]. Ischemic injury activates microglia and causes them to gather at the injury site and penumbra. These activated microglia release cytokines, chemokines, and growth factors to produce ROS, nitrogen species, and matrix metalloproteinase [12]. Microglia may prove detrimental after activation; however, they may be beneficial in some cases. Activation of inflammatory events after ischemia is responsible for blood–brain barrier damage. This damage causes fluid infiltration and edema. It also activates the release of free radicals. Therefore, it exacerbates post-stroke damage. Post-stroke inflammation ultimately promotes either recovery or cell death after triggering a cascade of immune cells [15].

Blood supply to white matter is lower than gray matter. Therefore, it is more prone to damage by ischemia [12]. Ischemia in white matter damages the axon and myelin sheath by activating proteases [12]. Excitotoxicity, peri-infarct depolarization, inflammation, and apoptosis are responsible for damage to tissues following ischemic stroke and are, therefore, considered potential targets for therapy.

The pathophysiology of cerebral ischemia is explained by the essential idea of excitotoxicity by a buildup of excitotoxic amino acids, including glutamate. This results in elevations in intracellular calcium and stimulates a number of signaling pathways, ultimately resulting in cell death. Additionally, energy-dependent cell pumps malfunction quickly after cerebral blood flow is slowed or stops because of the decreased glucose-dependent ATP synthesis, causing a flow of several ionic species inside the cell [16,17,18,19]. This results in cellular enlargement. Following ischemia, glutamate builds up in the extracellular environment and activates its receptors. Changes in the concentration of intracellular ions, most notably Ca2+ and Na+, are brought on by glutamate receptor activation [20,21,22].

## 3. FDA-Approved Drugs for Stroke

The actual goal of therapy is to reduce the risk of neuronal damage and long-term disability associated with ischemic stroke [23]. Tissue plasminogen activator (tPA) is currently the only FDA-approved thrombolytic for the treatment of acute ischemic stroke. tPA is an enzyme that can dissolve blood clots by digesting peptide bonds in proteins. It can convert plasminogen into plasmin. Plasmin is an enzyme capable of dissolving blood clots through the breakdown of fibrin molecules [24].

Synthetic tissue plasminogen activator can be manufactured using recombinant technology as is known as recombinant tissue plasminogen activator (rtPA). rtPA enzymes include alteplase, reteplase, and tenecteplase. Their half-life is prolonged and is more fibrin-specific [24]. rtPA has the propensity to dissolve the clot and restore blood flow in the blocked vessel. However, this drug is time-dependent, i.e., beneficial only if cell death has not yet occurred in the ischemic area. Therefore, this drug is only useful if administered quickly as the progression of necrosis is very fast in the ischemic area. Only through timely administration of rtPA, i.e., within 3–4.5 hours after a stroke attack, can salvageable penumbra be retrieved. Due to this time limit, only a few patients benefit from the treatment. Therefore, treatment aimed at saving patients from prolonged disability and death is difficult to attain [24].

## 4. Promising Phytocompounds for Stroke

### 4.1. Vitexin

Vitexin is a plant-derived phytochemical (Figure 2) present in many medicinal, and several other non-medicinal, plants, such as the passion flower, bamboo leaves, hawthorn, and chasteberry [25]. Chemically vitexin is a flavone C-glycoside (apigenin-8-C-β-D-glucopyranoside) and has been utilized in many studies to evaluate its potential in the treatment of stroke (Figure 2). These studies have shown that vitexin causes the inhibition of MCAO-induced (Table 1) brain infarction and apoptosis in animal models. Moreover, vitexin also causes a reduction in the levels of certain mediators, such as LDH, MDA, and NO, that are usually elevated due to increased levels of MCAO [19]. The enhanced inflammatory response in MCAO was also reduced when tested on rat models. The probable mechanism was by regulating the release of pro-inflammatory cytokines, such as TNF-α and IL-6, and anti-inflammatory cytokines, such as IL-10. According to the results, it is suggested that vitexin has to potential to suppress autophagy dysfunction to reduce MCAO-induced cerebral ischemic stroke through the TOR/U1k1 pathway [26].

### 4.2. Eriodictyol

Eriodictyol is a phytochemical (Figure 2) present in several fruits, vegetables, and certain Chinese herbs. It was first isolated from a Chinese herb called *Dracocephalum rupestre*. In a study, it was observed that the oral administration of eriodictyol in the doses of 1 mg/kg, 2 mg/kg, and 4 mg/kg has the potential to inhibit the effects of MPO expression and inflammatory cascade in a rat model of permanent ischemic stroke. It is suggested that eriodictyol treatment causes the down-regulation of the expression of certain inflammatory mediators, such as iNOS and TNF-α in ischemic cortex, which leads to a reduction in infarct size, improvement in motor function, and reduction in the memory deficit in rat models of permanent ischemic stroke [28]. Eriodictyol-7-O-glucoside (Figure 2) in 30mg/kg doses has also shown some neuroprotection against cerebral ischemic injury via activation of the antioxidant signalling nuclear factors, such as the erythroid-2-related factor 2/antioxidant response element (Nrf2/ARE) [29].

### 4.3. Carveol 

Carveol is a naturally occurring phytochemical of monocyclic monoterpenoid nature abundantly present in nature in several plants, such as caraway seeds, essential oils of orange peel, dill, blackcurrant berries, and black tea [50,51]. In a study, it was observed that carveol has the potential to activate the endogenous master anti-oxidant Nrf2, which ultimately causes the regulation of the expression of downstream anti-oxidants. This eventually led to the reduction in MCAO-induced neuro-inflammation and neurodegeneration in rat models [30].

### 4.4. Ferulic Acid

Ferulic acid is a biologically active phytochemical (Figure 2) present in many medicinal plants, such as Ferula foetida (asafoetida) and LC. In a study, it was observed that ferulic acid has superoxide radical scavenging activity and the potential to restore vasodilation in spontaneously hypertensive rat models [31]. Ferulic acid treatment in 100 mg/kg doses at the very start of the MCAO-induced stroke model has shown a reduction in infarct size and improvement in neurological deficits. The probable mechanisms were related to the effective inhibition of ICAM-1 and NF-kB and by decreasing the infiltration of MPO immune reactive cells [52].

### 4.5. Rosamirinic Acid

Rosamirinic acid (RA) is a naturally occurring phenolic phytochemical (Figure 2) present in many herbs such as rosemary and has been isolated from the Chinese herb *Salvia miltiorrhiza* [33]. In a study, it was revealed that RA treatment in 50 mg/kg doses causes the inhibition of HMGB1 expression and nuclear factor-kB activation, which helps in the reduction in BBB permeability and infarct size, and in the improvement of neurological deficits in rat models of cerebral ischemia-reperfusion injury [35].

### 4.6. Paeoniflorin

Paeoniflorin is a naturally occurring phytochemical (Figure 2) derived from certain varieties of Paeonia species, such as *Paeinia lactiflora Pall.* In a study, it was observed that paeoniflorin possesses anti-inflammatory properties. It was revealed that an IV administration of paeoniflorin at the doses of 10mg/kg, 15mg/kg, and 20mg/kg doses, 10 min before or 30 min after the MCAO-induced cerebral ischemia causes a reduction in infarct size and neurological deficits through inhibition of PMN infiltration and by the down-regulation of inflammatory factors, such as TNF-α, IL-1β, and ICAM-1 [53].

### 4.7. Allicin

Allicin is a sulphur-containing phytochemical (Figure 2) present in garlic species, such as *Allium sativum* [54]. Allicin is known to possess anti-inflammatory, antifungal, antioxidant, and anti-tumor properties [37]. In a study, it was revealed that in 50 mg/kg doses, allicin causes a reduction in the TNF-α levels and MPO activity. By inhibiting these inflammatory mediators, Allicin causes a reduction in infarct size and decreases brain edema. The authors also observed an improved neurological score in the experimental rat models of MCAO ischemic stroke [55].

### 4.8. Curcumin

Curcumin is a naturally occurring polyphenolic phytochemical (Figure 2) present in large quantities in the curry spice turmeric (*Curcuma longa*). It has been used for the treatment of several ailments in ancient Chinese and Indian medicine. In several studies, it was revealed that curcumin shows its effects by targeting multiple molecular targets, such as transcription factors and growth factors, along with their receptors [56]. Curcumin also possesses anti-inflammatory, antioxidant, anti-tumor, and cardiovascular-protective activities [57]. In 150–200 mg/kg doses, curcumin also possesses neuroprotective properties, which are due to the inhibition of leukocyte infiltration, regulation of microglia/macrophage polarization, and a reduction in the production of several inflammatory factors. Curcumin also inhibits autophagy against cerebral ischemia-reperfusion injury [39,40]. The inhibition of signalling pathways, such as TLR2/4-NF-kB, causes a reduction in MPO activity in ischemic stroke models [58]. 

### 4.9. Ginkgolide K

Ginkgolide K (GK) is a naturally-occurring phytochemical (Figure 2) isolated from the leaves of various Ginkgo species, such as *Ginkgo biloba* [42]. Ginkgolide K has been utilized as a traditional medicine in Chinese medicine for the alleviation of symptoms related to cerebrovascular and cardiovascular disorders. In an experimental study, it was observed that ginkgolides are the natural antagonists of platelet-activating factor (PAF) receptors. It possesses neuroprotective properties due to its potential to reduce inflammation and oxidative stress after ischemic reperfusion injury. In recent years, Ginkgolide has caught the eye of many researchers worldwide owing to its potential in the treatment of stroke and cardioprotective properties against endoplasmic reticulum stress injury after an episode of ischemia [42]. It is suggested that GK possesses a dual role in neuronal protection after ischemic reperfusion injury through the inhibition of mitochondrial fission caused by the phosphorylation of Drp1 (Ser637) and reducing mPTP opening in a GSK-3β-dependent manner in neurons. The available data suggests that Drp1 and GSK-3β might be important factors in the opening of mPTP after cerebral ischemic reperfusion injury [59].

### 4.10. 6″-O-succinylapigenin

6″-*O*-succinylapigenin is a phytochemical (Figure 2) which is generally derived from apigenin and present in many different herbs. Zhang et al. observed that the administration of 6″-*O*-succinylapigenin in MCAO-induced stroke rats causes a reduction in infarct volume and also causes an improvement in many neurological deficits. Moreover, it also possesses anti-oxidant properties and when administered in rat models, it causes the increase in SOD and HO-1 levels and a reduction in MDA. It also possesses the ability to restore the levels of phosphorylated ERK/ERK [44].

### 4.11. Forsythiaside A 

Forsythiaside A (FA) is a biologically-active phytochemical (Figure 2) present in large quantities in the fruits of *Forsythia suspense* [60]. In a study, it was found that FA possesses protective action in MCAO-induced rat models. FA possesses anti-oxidant properties by increasing the protein levels of Nrf2, NQO1, and GST, due to which, when administered in MCAO rats, it increased the survival rates by decreasing neurological deficits and apoptosis. FA has also been found to cause a reduction in serum MDA, an increase in SOD and GSH levels, and a reduction in endoplasmic reticulum stress [60].

### 4.12. Isoquercetin

Isoquercetin (Iso) is a phytochemical (Figure 2) present in various medicinal, as well as dietary plants, including several fruits and vegetables, and their derived drinks [45]. Dai et al. studied the protective mechanisms of Iso in ischemic reperfusion-injured rat models. Iso, when administered to MCAO-induced rats, causes a reduction in infarct size and cerebral edema. The results were more pronounced in rats that received the highest tested dose of Iso. Iso also improved the neurological score. In a study, it was also found that Iso possesses antioxidant properties and exerts its antioxidant action by causing a reduction in ROS and MDA levels and an increase in the levels of SOD and CAT in the hippocampi of MCAO-induced rat models. They also found that treatment with Iso also causes a reduction in apoptosis and increases the translocation of Nrf2 into the nucleus of the cell, thereby inhibiting the NOX4/ROS/NF-kB pathway. The in vitro analysis showed the same results when hippocampal neurons were exposed to OGD. However, it was observed that when the Nrf gene was knocked down, it suppressed the protective effects of Iso [47].

### 4.13. Trilobatin

Trilobatin (TLB) is a plant-derived phytochemical (Figure 2) and one of the main constituents of *Lithocarpus polystachyus.* In an experimental study, Gao et al. observed that it helps in improving neurological deficits, and brain edema and reducing infarct volume in MCAO-induced rat models. Moreover, they also estimated the time window for the administration of TLB after MCAO, which indicated that TLB in doses of 20mg/kg exerted a neuroprotective action after MCAO within the 4 hours. However, the protective effects were lessened when administered at 6 hours after MCAO. It was also found that TLB causes the restoration of long-term neurological functions. The neuroprotective effects of TLB were due to its ability to suppress the inflammatory cascade, which was evidently seen by the reduction in astrocyte and microglia activation, and decreased levels of pro-inflammatory markers. TLB also suppressed the levels of ROS and MDA and increased the activities of SOD and GPx. The antioxidant action of TLB was also associated with the upregulation of several nuclear factors, such as Nrf2, NQO1, and HO-1 expression, together with the suppression of Keap1. TLB also causes an elevation in the expression of Sirt3. In vitro replicative studies also showed the same results [61].

### 4.14. Genistein

Genistein (Gen) is a naturally-occurring isoflavone phytoestrogen (Figure 2) found in great proportion in soy foods. Several studies have shown that Gen has the potential to lessen the harm caused by both focal and global cerebral ischemia (GCI) in female mice with ovariectomy [62]. It is suggested that Gen treatment may reduce acute injury caused by cerebral ischemia in reproductively senescent mice through the inhibition of NLRP3 inflammasome present in microglia. This suggested that Gen may be a drug candidate for treatment of stroke in postmenopausal women [48].

### 4.15. Tocotrienol

Alpha-tocotrienol (Natural vitamin E) occurs in palm oil, wheat germ, barley, etc (Figure 2). It provides protection against ischemic stroke by induction of MRP1. The intracellular outflow of oxidised glutathione from brain cells is mostly mediated by MRP1. This is due to the potential for brain-cell death caused by an increase in intracellular oxidised glutathione. Tocotrienol can also promote neuroprotection by controlling the expression of microRNA in stroke-affected brain tissue. It has been demonstrated that vitamin E helps prevent ischemic stroke. However, it also raises the risk of subarachnoid haemorrhage [63,64,65].

## 5. Conclusions and Future Direction

This study aims to provide a thorough literature search of phytocompounds that might be useful in the treatment of stroke. Since ancient times, phytocompounds have been utilised to treat neurological illnesses. These molecules have shown to be quite useful even today, particularly in the discovery of new lead compounds that target neurodegenerative illnesses. Extensive experimental research on these plants has led to the discovery of several phytocompounds whose effects have been shown to go well beyond just attenuating stroke. After a thorough literature search, we have concluded that a handful of compounds that can play a role in the attenuation of stroke. Further changes to these compounds after structure-activity relationship studies might aid in the discovery of safer phytocompounds for stroke therapy. Future research should therefore focus on finding new lead drugs with improved effectiveness and a broader therapeutic window.

## Figures and Tables

**Figure 1 biomedicines-10-02566-f001:**
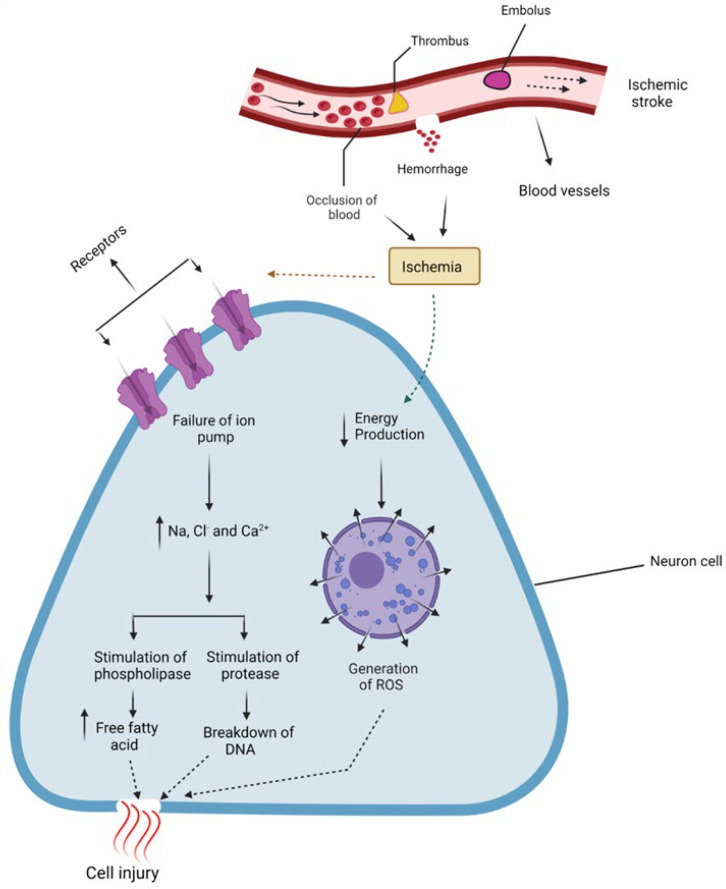
Pictorial presentation of ischemic stroke, the consequence of reduced blood supply due to occlusion of the main cerebral artery, either temporarily or permanently. In this way, the central core becomes devoid of blood supply, and energy-dependent cell pumps malfunction, causing a build-up of Na+ and Ca+ and decreased energy production leading to the generation of Reactive Oxygen Species (ROS), and resulting in neuronal damage.

**Figure 2 biomedicines-10-02566-f002:**
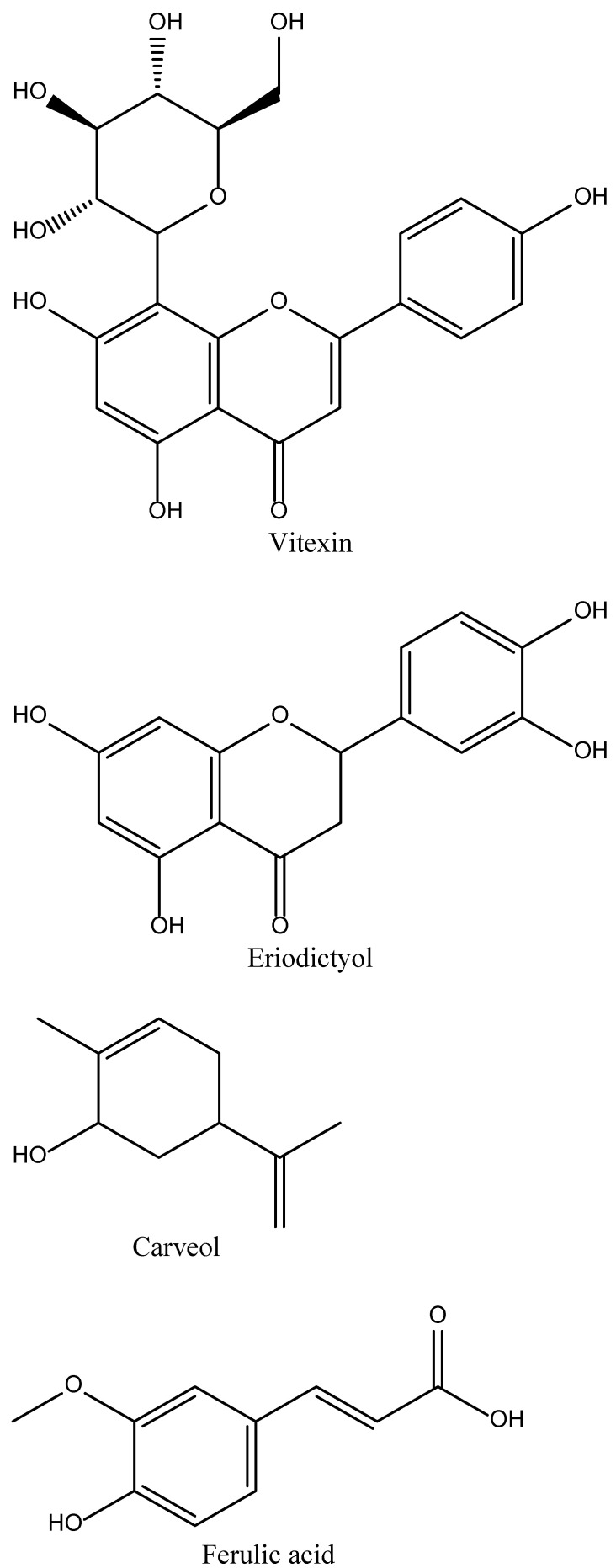
Promising phytocompound’s structures for stroke.

**Table 1 biomedicines-10-02566-t001:** Phytocompounds, their biological source, nature of the compound, the dose given to animal models, route of administration, oral bioavailability, and animal model.

Name of Natural Compound	Botanical Source	Nature of Compound	Dose	Route of Administration	Oral Bioavailability	Animal Model	References
**Vitexin**	*Vitux agnus-castus*, *Vitex negundo*	Flavone C-glycoside (apigenin-8-C-β-D-glucopyranoside)	2 mg/kg	Intravenous	Poor due to very high 1st pass metabolism	MCAO induced Rats stroke model	[25,27]
**Eriodictyol**	*Dracocephalum rupestre*	Flavonoid	1,2 and 4 mg/kg	Oral	Good	Male Swiss mice model and Rat model of focal cerebral ischemia	[26,28]
**Carveol**	Caraway seeds (*Carum carvi*), Dill seeds, Spearmint seed	Monocyclic monoterpenoid	20 mg/kg	Intraperitoneal	Good	MCAO-induced ischemic stroke model	[29,30]
**Ferulic acid**	*Ferula foetida* (asafoetida)	Phenolic phytochemical	80 and 100 mg/kg	Intravenous	Highly variable (0.4–98%)	MCAO-induced stroke model	[31,32]
**Rosamirinic acid**	*Salvia miltiorrhia*	Polyphenol	50 mg/kg	Intravenous	Extremely poor (1.57%)	Rat models of cerebral ischemia-reperfusion injury	[33,34]
**Paeoniflorin**	*Paeinia lactiflora* Pall.	Monoterpene glycoside	20 mg/kg	Intravenous	Very poor (2.32%)	MCAO ischemic stroke rat model	[35,36]
**Allicin**	Garlic species such as *Allium sativum*	Organosulfur compounds	50 mg/kg	Intraperitoneal	18%	MCAO ischemic stroke rat model	[37,38]
**Curcumin**	*Curcuma longa* (turmeric)	Diarylheptanoid	300 mg/kg 50 mg/kg	Intraperitoneal	Very poor	MCAO ischemic stroke rat model	[39,40,41]
**Ginkgolide K**	*Gingko biloba*	Terpenoids	2.4 and 8 mg/kg	Intraperitoneal	Approximately 80%	Rat Ischemia model	[42]
**6″-O-succinylapigenin**	*Cynara cardunculus var. scolymus* L	Flavonoid	20, 40 and 60 mg/kg	Intraperitoneal	Good	MCAO-induced rat models	[43]
**Forsythiaside A**	Fruit of *Forsythia suspense.*	Hydroxycinnamic acid	50 mg/kg	Intraperitoneal	Moderate	MCAO-induced rat models	[44]
**Isoquercetin**	Citrus fruits, apples, onions, parsley, sage, tea, and red wine. Olive oil, grapes, dark cherries	O-Glucoside	20 mg/kg	Intravenous	Moderate (51%)	MCAO-induced rat models	[45,46]
**Trilobatin**	*Lithocarpus polystachyus*	Polyphenol	20 mg/kg	Intravenous	--	MCAO-induced rat models	[47]
**Genistein**	Plants including lupin, fava beans, soybeans, kudzu,	7-hydroxyisoflavone	10 mg/kg	Intraperitoneal	23.4%		[48,49]

## Data Availability

Not applicable.

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
