# Peer review of "A Review on Therapeutic Potential of Natural Phytocompounds for Stroke"

_biomedicines, 2022, doi:10.3390/biomedicines10102566_

Round 1
Reviewer 1 Report
Peer-review of the manuscript: ”A review on therapeutic potential of natural phytocompounds for stroke” (Manuscript ID: biomedicines-1940556) – submitted for publication in the Journal Biomedicines (ISSN 2227-9059)j
1. General remarks:
The authors achieved a quite useful – as long as, unfortunately, to date there is (still !) no therapeutic (pharmacological, surgical physiatric) agent/ intervention able to effectively cure the central nervous system’ lesions, inclusively of the brain after ”completed” (MacDonald BK, Cockerell OC, Sander JW, Shorvon SD. The incidence and lifetime prevalence of neurological disorders in a prospective community-based study in the UK. Brain. 2000 Apr;123 (Pt 4):665-76. doi: 10.1093/brain/123.4.665. PMID: 10733998.) stroke – narrative review ”on therapeutic potential of natural phytocompounds for stroke”.
2. Specific remarks:
- the suggestive figure at pag. 3 does not have neither number nor any explaining text below, and additionally: is it original or reroduced – a nd if so, after who, and is it with the author(s)’ permision ?
- the role of excitatory aminoacids and their receptors post stroke (Onose G, Anghelescu A, Blendea D, Ciobanu V, Daia C, Firan FC, Oprea M, Spinu A, Popescu C, Ionescu A, Busnatu Ș, Munteanu C. – Cellular and Molecular Targets for Non-Invasive, Non-Pharmacological Therapeutic/Rehabilitative Interventions in Acute Ischemic Stroke. Int J Mol Sci. 2022 Jan 14;23(2):907. doi: 10.3390/ijms23020907. PMID: 35055089; PMCID: PMC8846361.) is not specifically approached in the ”Pathophysiology” section of this manuscript
- ”Secondary damage to the immune system clears the damaged tissues”: this assertion used by the authors should be clearified
- ”This study aims to give a complete overview of phytocompounds ...”: as long as there is not a systematic review – this kind of work being a largely accepted, structured concept for fulfilling a query over a targeted knowledge area, and using a standardized related method, including with steps of the identified papers’ filtering/ selection (for instance: PRISMA) – the formulation used, i. e. ”complete overview” is questionable. Therefore, the authors should present in more cautious terms, their endeavor, than ”thorough literature search”.
Author Response
Dear Respectable Reviewer please find enclosed here with author response as per your recommendation.
regards

Reviewer 2 Report
A review written by Farooq et al. on therapeutic potential of natural phytocompounds for stroke is interesting and informative. The authors discuss the role of natural products against stroke. The review article can be accepted for publication after minor revision.
The structures of the compounds need to be refined and better to draw the compounds in ChemDraw.
2. Page 4, line 155, in a study it was observed……. In which study it was observed, there is no reference cited.
3. I think better to add one paragraph about the reported medicinal and traditional activities of these isolated compounds with proper isolation sources from where these compounds are isolated in a good amount.
4. Page 7, line 226, there is no reference available for the isolation of curcumin from curcuma longa. Curcuma longa must be italic. Page 217, no reference for Allicin from Allium sativum.
Change the sentence “Ginkgolide K (GK) is a naturally occurring phytochemical which is isolated from the leaves of various Ginkgo species such as Ginkgo biloba” in to “Ginkgolide K (GK) is a naturally occurring phytochemical isolated from the leaves of various Ginkgo species such as Ginkgo biloba. Add a reference also.
6. Page 8, Line 267, cite the reference at the end of the sentence. ………..present in the fruits of Forsythia suspense.
The review recently published by Zhang B, Saatman KE, Chen L. Therapeutic potential of natural compounds from Chinese medicine in acute and subacute phases of ischemic stroke. Neural Regeneration Research. 2020 Mar;15(3):416. Is mostly related to the work but did not cite.
Some of the references in reference section are not formatted according to the format of the journal.
9. Most of the journal names in references are not abbreviated and italic. Need to be corrected as per journal format.
Author Response
Dear Respectable Reviewer, please find enclosed here with author response as per your kind suggestion.
regards
Dr. Aman Ullah

Reviewer 3 Report
Manuscript ID
biomedicines-1940556
Type: Review
Title: A review on therapeutic potential of natural phytocompounds for stroke
The conceptualization is sound but I am afraid this review is poorly executed.
The discussion is not in-depth and appeared to focus on animal models rather than on human.
The cited examples seem not being critically analysed and not able to form a conclusive recommend on whenever its therapeutic potential is true or noteworthy
The figure 1 does not have caption and the sequence of the cascade is not given
The type of phytocompounds shown also not exhaustive, tocotrienol should be included too
Author Response
Dear respectable reviewer please find enclosed here with author response as per your kind suggestions.
regards
Dr. Aman Ullah

Round 2
Reviewer 1 Report
I have already sent them on my first revision and the authors considered and implemented my suggestions for optimization of this manuscript.
Reviewer 3 Report
No further comment